# JMAC Protocol: A Cross-Layer Multi-Hop Protocol for LoRa

**DOI:** 10.3390/s20236893

**Published:** 2020-12-02

**Authors:** Juan José López Escobar, Felipe Gil-Castiñeira, Rebeca P. Díaz Redondo

**Affiliations:** AttlanTTIC Research Center, University of Vigo, 36310 Vigo, Spain; rebeca@det.uvigo.es

**Keywords:** smart city, IoT, LoRa, multi-hop, mesh network, OMNeT++

## Abstract

The emergence of Low-Power Wide-Area Network (LPWAN) technologies allowed the development of revolutionary Internet Of Things (IoT) applications covering large areas with thousands of devices. However, connectivity may be a challenge for non-line-of-sight indoor operation or for areas without good coverage. Technologies such as LoRa and Sigfox allow connectivity for up to 50,000 devices per cell, several devices that may be exceeded in many scenarios. To deal with these problems, this paper introduces a new multi-hop protocol, called JMAC, designed for improving long range wireless communication networks that may support monitoring in scenarios such smart cities or Industry 4.0. JMAC uses the LoRa radio technology to keep low consumption and extend coverage area, and exploits the potential mesh behaviour of wireless networks to improve coverage and increase the number of supported devices per cell. JMAC is based on predictive wake-up to reach long lifetime on sensor devices. Our proposal was validated using the OMNeT++ simulator to analyze how it performs under different conditions with promising results.

## 1. Introduction

Since the development of the Internet of Things (IoT) paradigm a few years ago, related technologies have dramatically evolved to turn IoT into a widely spread reality. The evolution in Low-Power Wide-Area Network (LPWAN) technologies [1] is particularly remarkable, which support long range communications with notably low power consumption, becoming a good alternative to traditional wireless networks that do not support these communication requirements: (i) Wireless Wide Area Networks (WWAN) support long distance communications, but require from higher power consumption; and (ii) Wireless Personal Area Networks (WPAN) require from lower power consumption, but they are not suitable for long distances. In fact, there is a general consensus about what characteristics are desirable in LPWAN [2]: (i) low power consumption, needed for batteries; (ii) inexpensive chips; (iii) easy and scalable deployment; (iv) ALOHA with single hop routing as Medium Access Control (MAC) protocol; (v) secured data; and (vi) robust radio modulation to minimize the impact of channel fading.

There are some relevant technologies within the LPWAN field. LoRaWAN [3], a point-to-multipoint networking protocol that is built upon the LoRa physical layer [4], must be highlighted for different reasons, being its easy deployment and open nature two of the most remarkable ones. Additionally, Sigfox [5] is managed by a single operator and offers a complete IoT architecture, although quite constrained because of using a proprietary UNB modulation and infrastructure. Besides, DASH7 [6] and Weightless [7] are open standardized stacks, but with lower impact in the IoT domain. All the above options have in common the use of unlicensed bands of the spectrum, but there are alternatives in licensed bands from the 3rd Generation Partnership Project (3GPP) [8]. This is the case of NB-IoT [9], which exploits the existing cellular network to connect devices which generate a small data flow, or LTE-M [10], a wider bandwidth option for applications which frequently require to exchange high volumes of data.

Most of these technologies are currently used in many situations and support an only single hop communication. This means that for large areas it is necessary to deploy infrastructure for unlicensed technologies, or to get a subscription to a network operator. This problem is exacerbated in areas where physical phenomena (interference, noise, obstacles, etc.) reduce the coverage range to a few kilometers or hectometers, such as in urban areas or in industrial settings (indoor places). Our proposal precisely tries to overcome this problem, but still satisfying the LPWAN general requirements previously mentioned. Thus, our first contribution is to present an innovative layout for multi-hop networking on top of a meshed version of LoRa communication, which we coined as JMAC. Enabling multi-hop capability allows reusing resources and expand the coverage area substantially. Our second contribution is the development of an appropriate framework within the OMNeT++ simulator [11] that behaves as close as possible to the real behaviour of LoRa, which we coined as FLoRaPHY. We needed from this new software module to check the behaviour and scalability of the new protocol. This software is available for the research community in GitHub [12].

This paper is organized as follows. Section 2 describes the base technologies that we used to define our new protocol: the LoRa technology, needed for the physical layer and that supports the new protocol JMAC and the OMNeT++ simulator, used for validation purposes. Section 3 summarizes the related work within the multi-hop networks, not only using LoRa and LoRaWAN, but also describing interesting approaches for multi-hop Wireless Sensor Networks (WSN). Section 4 describes the JMAC protocol: behaviour, frame formats and design restrictions. In Section 5 we describe the new software module created to simulate LoRa within the OMNeT++ simulator and we also show the results obtained under two different scenarios: a simple one, to check the impact of the different parameters in the protocol, and a more complex and dense topology to check the scalability of the proposed protocol. Section 6 summarizes conclusions and limitations of our proposal, giving some clues to further improvement. Finally, Section 7 summarizes the conclusions and future work.

## 2. Background

The JMAC protocol works with LoRa, a lower physical layer described in Section 2.1, to offer a suitable multi-hop solution or the upper networking layer. Usually, solutions over LoRa work with LoRaWAN, a protocol defined precisely for this networking layer, so this approach is also explained in the same subsection. Since we need to validate our proposal, we opted to use the OMNeT++ simulation environment, which is described in Section 2.2.

### 2.1. LoRa

The origins of LoRa date back to 2010, as a new long range and low power modulation technique created by the French start-up Cycleo. Two years later, the American company Semtech acquired Cycleo and (i) the first LoRa chips for end-devices and gateways were launched to market and (ii) the new protocol LoRaMAC was created under a proprietary license in order to enable networking into the upper layers and define message format and security aspects. In 2015 the LoRa Alliance [13], an open, nonprofit association of companies, universities, research groups and developer communities worldwide, was founded and the LoRaMAC protocol was renamed to LoRaWAN, with the aim of establishing a new standard for LPWAN. In fact, the LoRa Alliance is only in charge of writing the LoRaWAN specification of the technical implementation to ensure interoperability between manufacturers and developers, whereas companies are free to define any commercial model or type of deployment of cloud services (public, private, hybrid or enterprise). Notable examples of real world deployments are The Things Network (TTN) platform [14], an open cloud service which leads LoRaWAN usage and is supported by a great community, and its commercial version The Things Industries (TTI) [15].

The LoRa technology is based on a proprietary version of the Chirp Spread Spectrum CSS modulation [16], which tolerates receiving successfully signals below the noise floor. This increases the link budget and the immunity to interference, which allows long range communication and gives rise to the technology name. It is defined as a spread spectrum technique which symbols are transmitted using up- and down-chirps, i.e., a digital chirp signal that varies in frequency within boundaries of the channel. This simple concept has the advantage of having equivalent timing and frequency offsets between transmitter and receiver, simplifying the receiver design. Despite the limited information provided by the manufacturer in a few manuals [17,18], many studies have addressed the objective of reverse-engineer the exact modulation technique, experiment with it in real world and try to guess a mathematical model [19,20,21,22,23], so more information about it is available nowadays. LoRa was not the first modulation of its kind, since it is popular in radar applications. However, LoRa was the first low cost implementation for commercial usage, which gives the opportunity to exploit this kind of technology in new contexts, such as IoT and smart cities because of its main characteristics:Narrowband/Wideband: it can operate in the same way both narrowband and wideband configurations because bandwidth and central frequency are scalable and easy to adapt to any application requirement.Constant Envelope: the information of the signal lies in frequency variation and it is independent of the amplitude. Then, the low-power high-efficient power amplifier can operate at saturation level or near it.High Robustness: symbols are very long compared to bandwidth, which provides outstanding immunity to adjacent-channel interference.Pseudo-orthogonality: it might be one of the most relevant and interesting topics in LoRa. Spreading factors (SF) have an effect that allows transmitting/receiving correctly multiple signals in the channel at the same time, as long as they use different SF. This feature may be expanded to some combinations of overlapping channels as long as the power difference between them is high enough to consider the interference as noise, as it is shown in [21].Multipath/Fading Immunity: chirp pulses are relatively long time duration, therefore they are resistant against multipath and fading of the signal.Doppler Resistant: mobile communications are correctly supported by LoRa, since the Doppler effect only generates a small and negligible frequency shift in the pulse which does not require very accurate clock sources.Localization: LoRa is suitable for ranging transmitter location due to the ability to discriminate between frequency and time errors which may be produced by multi-path effects, similar to radar applications.

The LoRaWAN architecture is deployed in a star-of-stars topology (Figure 1) in which single-channel end-devices communicate directly with any multi-channel gateway. Gateways are connected to a back-end through conventional Internet, and serve as a transparent bridge between end-devices and back-end, so they must be active at all times. In this architecture, end-devices are usually sensors that send data eventually and try to waste as little energy as possible. They are usually classified into three types. (i) Class A devices send an uplink message at any time and stay in receiver mode for two short downlink windows right after, to give the opportunity for bidirectional communication or for receiving control commands. The rest of the time stay in sleep mode, so they have the lowest power operating mode. This is the default class and must be supported by all LoRaWAN end-devices, as it is used to build the following ones. (ii) Class B devices operate as Class A, but they add an extra receiving window periodically. This window is announced via beacon frames to time-synchronize the back-end. Finally, (iii) Class C devices keep the receiving window open at all times, except when transmitting (half-duplex), so the latency for downlink messages is decreased drastically. Such devices should not operate with batteries because of the high power consumption. The class of end-devices actually determines the MAC protocol used in LoRaWAN.

Gateways must implement a Packet Forward layer to convert LoRa RF packets to standard IP packets and vice versa to allow bidirectional communication. In addition, they are usually able to operate simultaneously in combinations of frequency channel and SF, which enhances network capacity and allows frequency hopping.

The back-end is an infrastructure composed of several micro-services detailed in [24]. It is essentially comprised of three units. (i) The Network Server (NS) that is the center of the star topology, marks the edges of the LoRaWAN MAC layer for the end-devices and processes and routes the messages. It is responsible for data rate adaptation, MAC layer issues (control and commands) and queuing downlink messages. Besides, NS copes with roaming aspects if it were necessary. (ii) The Join Server (JS) manages authentication of end-devices and activation of keys via Over-The-Air (OTA) procedure. It stores all necessary elements to identify securely an end-device and must establish a secure communication with NS. Finally, (iii) The Application Server (AS) handles all application features, both managing payload uplink and downlink messages, offering this data to the end-user. It must share with the Join Server (JS) the application credentials.

The LoRaWAN specification considers spectrum usage in different regulatory regions worldwide [25] and identifies the most common channel plans, although we focus on the most common in European countries (EU868-870). Specifically, it must operate at least in the 3 default channels (868.1 MHz., 868.3 MHz. and 868.5 MHz.) with a restriction of at most 1% duty cycle. Currently, the most popular version is v1.0.3 [26], but the latest one is v1.1 [27], which adds new functionalities.

### 2.2. OMNeT++

OMNeT++ [11] is a popular open-source discrete event simulator for any kind of network, supporting from wired and wireless networks to photonic ones. It provides a complete extensible and modular framework written in C++ and an Eclipse-based IDE with a graphical environment to facilitate development, simulation execution and analysis of data results and performance of the network. It is distributed under an Academic Public License which allows using it in non-commercial environments, for instance research and teaching. There is also a commercial version called OMNEST [28]. Besides, it is supported by an important community, making it a very suitable option for modelling computer networks scenarios. OMNeT++ defines a lightweight class called cObject as root element for any simulation functionality. In order to represent the simulation scenario, the cModule and cChannel classes are defined. The former deals with the main logic of the modules, whereas the latter is in charge of modelling the physical medium. In addition to the kernel libraries, projects are built using a topology description language, called NED, which allows assembling complex components and reuse modules using a high-level language.

One of the strengths of OMNeT++ is the amount of external frameworks that have been developed by the community, whether they are researchers or independent developers. The simulation models are indexed in the official website, but not all of them are completely mature or offer technical support. However, there is one model library which stands above the others: the INET Framework [29]. It contains modules for any traditional Internet protocol (TCP, UDP, IPv4, IPv6, BGP, Ethernet, IEEE 802.11, etc.), physical modulation techniques (APSK, DSSS, etc.), mobility and QoS support, power consumption estimators and a lot more which may be useful to design and evaluate new technologies. Additionally, INET is often the basis for further libraries, since it provides essential components which fit perfectly and save time in development phase, such as vehicular networks or LTE.

Aside from versatility of the simulator, the main reason it was chosen was the availability of a LoRaWAN-oriented framework, called FLoRa [30], developed by researchers from the Aalto University (Finland). It does not support the complete LoRaWAN protocol, but the crucial parts to implement new simulation scenarios, such as the developed in this work, are there and can be used.

## 3. Related Work

Sensor network deployments, also LoRaWAN networks, may have coverage capacity problems when they are used in adverse scenarios, such as in urban or indoor areas, due to high interference, noise level and path losses [31]. Consequently, in the last few years, researchers have designed mechanisms to create multi-hop networks that can expand the coverage under these circumstances. These proposals can be organized into three philosophies: (i) providing a completely new protocol stack over the LoRa radio (Section 3.1); (ii) extending a LoRaWAN network with multi-hop routing between end-devices or between intermediate gateways (Section 3.2); and (iii) using WSN approaches that are more mature in the multi-hop strategies field (Section 3.3).

### 3.1. LoRa Multi-Hop

A very clear trend in mesh networks using LoRa technology is exploiting the features of LoRa modulation to implement the Concurrent Transmission (CT) protocol [32], as in [33]. Concretely, the CT-based multi-hop protocol has an initiator which broadcasts a message along the network using the other nodes as flooding forwarders. This is possible because the LoRa modules only receive correctly the highest-energy packet and no collision avoidance mechanism has to be considered. In order to support all nodes to initiate the transmission, they must be synchronized by a Time-Division Multiple Access (TDMA) protocol using the received messages. Moreover, this approach is improved with a small timing offset insertion during the transmission time, which leads to a better receiving performance.

Another relevant proposal is the deployment of 19 LoRa mesh nodes in the campus of the National Chung-Cheng University (Taiwan) which implement a multi-hop network coordinated by a central gateway which requests sequentially the data from each sensor [34]. It is clear that the main drawback of this proposal is the fact that the medium access is managed by the gateway, thus they have been working more on this project and developed a mechanism to send emergency packets automatically from the nodes [35].

One of the first attempts to define multi-hop networks using LoRa is LoRaBlink [36]. They propose a TDMA protocol that assumes the communication is only between the nodes and a sink, which uses beacons to synchronize the nodes in each epoch, and it benefits from the concurrent transmission feature of LoRa to decode correctly at most only one packet in each slot and receiver.

In [37] the authors present an anycast LoRa multi-hop application, known as LOCATE, to enable emergency warnings where cellular connectivity is not available. This is achieved through a DTN dissemination mechanism which tries to avoid collisions randomly.

Additionally, a linear multi-hop LoRa network is implemented inside the aqueducts of Siena (Italy) [38]. This system was designed under the assumptions of an underground environment which is free of collisions. This ensures that the presented synchronization algorithm can optimize the wake-up time of the nodes.

Finally, a new TDMA scheme especially designed to achieve low-latency multi-hop over LoRa was introduced in [39]. It operates in three phases, an initial one to build the tree structure and assign the correct channel and time slot to the nodes. Next, periodic groups of several upward and one downward cycles are executed in accordance with the initialization setup. Despite positive points related to latency and reliability, a deep study about the energy consumption must be done to evaluate its suitability.

Regarding practical implementations, some efforts were made to standardize the LoRa stack using the open-source Contiki OS, an open-source operating system for resource constrained IoT devices, and take advantage of the included protocols for WSN such as Routing Protocol for Low-Power and Lossy Networks (RPL) and Radio Duty Cycling (RDC), as well as enabling IPv6 connectivity. This was first developed in [40] for a simple point-to-point link. Then, in [41] the authors empowered a small LoRa multi-hop network using a custom MAC layer using the fast loop technique to select the suitable SF (RLMAC) and the RPL protocol. Further pursuing the desire to standardize LoRa networks, a complete framework called KRATOS for Contiki OS over LoRa physical layer was developed and open to everybody [42]. Similarly, a new protocol stack called HARE [43] oriented towards the deployment of power-efficient uplink multi-hop wireless networks was implemented in Contiki OS. It takes advantage of the existing protocols in Contiki, such as TDMA and RPL, to create a protocol stack which is agnostic to the LPWAN technologies in the physical layer. Since the traditional Contiki OS is currently abandoned, a new one was released in 2017, Contiki-NG [44]. Precisely, in [45] the TCSH mechanism of the IEEE 802.15.4 standard is adapted to LoRa under this new IoT OS. This way, this solution allows creating LoRa mesh networks using the RPL routing algorithm which meet the duty cycle regulation.

Furthermore, the IoT company Pycom recently developed a MicroPython API which enables an easy implementation of LoRa mesh networks [46]. It was implemented using the OpenThread stack, an open-source implementation of Thread, a protocol stack designed by Google, which provides reliable communication and tries to standardize mesh networking technology. This solution is used in [47] to send the information of fire-detecting sensors to the backend.

### 3.2. LoRaWAN Multi-Hop

As previously mentioned, there are other approaches that tried to extend a LoRaWAN network with multi-hop routing. One alternative is providing multi-hop routing between end-devices, such as in [48] where a modified version of DSDV routing protocol is successfully implemented in a linear topology of nodes, although energy consumption is not considered. Besides, underground sensors are considered in [49] using a tree topology with TDMA synchronization for the receive slots, which reduces power consumption. Finally, [50] sets a variation of LoRaWAN out, indeed it keeps in mind the imbalance coverage of the gateways to establish the routing tables of the end-devices.

The other alternative, which has been widely exploited, is providing multi-hop routing between intermediate gateways, such as in [51] mixing the HWMP and AODV protocols to encapsulate the LoRaWAN packets between final and intermediate gateways. A further proposal defining a new LoRaWAN class to create mesh networks between gateways (both final and intermediate gateways), which employs estimated time-on-air as metric and packet aggregation and disaggregation to increase throughput, is explained in [52].

### 3.3. Multi-Hop WSN

The WSN field offers interesting alternatives for wireless multi-hopping. Although they are generally old designs, it is worthy to summarize their main characteristics, since they may inspire strategies for our purpose: create a LoRa based network with multi-hop capabilities.

Sensor MAC (S-MAC) [53] is considered one of the first medium access protocols for sensor networks. Inspired in the RST/CTS mechanism of the 802.11 standard, which provides collision avoidance and good scalability, it is the basis of many other ones. In S-MAC, nodes are programmed to sleep during long periods of time to save energy and control the duty cycle, so they have to disseminate their wake-up schedules to the neighbours to enable synchronization among transmission and reception. Given that idle listening is one of the main sources of energy waste, in Timeout MAC (T-MAC) [54] the active period ends earlier. Similarly, the Adaptive energy efficient MAC (AEEMAC) [55] proposes three optimization techniques to save frame slots by combining control packets and avoid overhearing using the information that has been sent to the channel.

The lightweight MAC (LMAC) [56] protocol is based on another strategy: a distributed TDMA synchronization. This is carried out indicating which time slots are occupied in the current broadcast coverage area, and assigning a free time slot on it randomly.

WiseMAC [57] combines Preamble Sampling technique with learning neighbours’ sampling schedule: nodes must sample the channel for a short time periodically to receive data, and must send a preamble before sending the data frame at the correct moment of the receiver. The information about the receiver wake-up schedule is computed from the control information in the ACK frames. In the case of the first communication to perform by a node, a long preamble is used to be detected by any neighbour.

B-MAC [58] obtains good performance by executing the Clear Channel Assessment (CCA) procedure, a CSMA-based mechanism to estimate noise floor using software automatic gain control, before transmitting. When the channel is empty, some preambles are sent to aware the receiver about the next data frame. Nodes are configured to stay asleep for long time periods and wake up periodically to check potential receptions.

PMAC [59] adopts a completely new scheme: each node shares with the neighbourhood its tentative sleep-awake pattern (string of bits) over several slot times, so they know the potential moments to be active. This activity pattern is generated according to the network traffic. The results stand out over traditional approaches, such as S-MAC, in terms of power consumption and throughput.

In the X-MAC [60] protocol the main aim is minimizing the energy waste in nodes that are listening to data frames that are not addressed to them. Thus, in X-MAC the transmitter sends a short strobed preamble with information about the receiver. Then, the target node acknowledges the reception the earliest possible using pauses between preambles, and the rest of nodes go back to sleep quickly. In addition, an optimal algorithm to control duty cycle is presented, resulting network traffic load adaptation.

In AS-MAC [61], nodes wake up periodically to receive data and the transmission of data is scheduled to the target node’s receiving window. Concretely, when a node has a packet to send, it waits for the right moment, when the receiver is awake. This moment might also be a “Hello time” when the receiver sends a Hello packet with information about scheduling and offset time before the data packet. Before the periodic listening-sleep phase, an initialization is performed to discover the direct topology and set the node up.

Finally, another valuable idea is introduced in Receiver-Initiated MAC (RI-MAC) [62], whose nodes wake up periodically and broadcast a beacon to notify any transmitter that they are ready to receive data. The data frames are acknowledged by another beacon which allows the same transmitter to send more data if necessary. Therefore, collision and overhearing are reduced, and it can operate over a wide range of traffic loads. Following up on this idea, PW-MAC [63] changes the fixed schedule to pseudo-random schedule which avoids neighbour nodes to wake up at the same time constantly.

### 3.4. Comparison

The available approaches for multi-hop might be organized into two main groups: (i) LoRa and LoRaWAN solutions and (ii) WSN protocols, whose main characteristics and performance results are summarized in Table 1 and Table 2 respectively. The column *Link Layers* displays the link protocols used to control the medium access and to synchronize the communications. Those solutions based on time division and channel activity detection must be highlighted since they achieve low consumption and good reliability, as well as protection against collisions. The column *Network Layers* shows the network protocol chosen to enable routing and topology shaping, whereas the column *Cross-Layer* shows if the proposal combines link and network layers in a single one. Since WSN protocols are mainly link layer ones, these two last columns were not included in Table 2. Regarding the performance, we completed an in-depth analysis of the publications to determine their relative Reliability, Consumption or Latency. Both tables include a column with a qualitative assessment of (i) *Reliability*, the degree of messages that are correctly delivered, (ii) *Consumption*, the amount of energy required for the proposal to properly work, (iii) *Latency*, the delay to reach the collector of the messages and (iv) the *Duty Cycle Limit*, which states if the restrictions on usage of frequency bands that are regulated by law are respected or not.

## 4. JMAC: A New Protocol

Our proposal is designed for sensing smart cities and, consequently, has to fulfil several characteristics. First, (i) we considered a cross-layer approach (combining link and network layers) to support a complete and flexible solution for different applications in smart cities. Second, and since sensors are powered by batteries, (ii) assuring low power consumption is essential. Since the main purpose is gathering information from an indefinite number of sensors of different nature to, at least, one sink or gateway, (iii) we propose to work under a tree-topology. Without losing flexibility, (iv) we assume the packet length is fixed by the APP layer, whereas (v) periodicity of generated data is considered to be limited to fifteen minutes at most. Additionally, (vi) the delay to deliver a packet to the final destination is unknown. Since downlink messages (from gateway to sensors) are mostly intended to conduct control operations, (vii) we consider they are rarely sent, so they must be prioritized (viii). We also assume that (ix) the sensing network is fixed, with sporadic changes is the topology (addition of new devices and/or shutoff of sensors). Finally, (x) since LoRa works on license free but regulated bands, the new protocol must comply the regulation relating maximum transmission power, radio frequency channel arrangement and duty cycle over the usage of the channel.

Consequently, our proposal, the JMAC protocol, broadly consists of a cross-layer architecture formed by a link layer similar to AS-MAC [61] and RI-MAC [62] protocols and a network layer inspired by a tree structure with meshed routing. It supports two types of devices. First, a *gateway* (or sink) that is the destination of the uplink messages. It serves as liaison with the final back-end and is the root node of the tree topology. It is always in receiver mode, except when it has to send control messages downwards the sensors in the network (half-duplex). There may be more than one gateway in the same network for redundancy and they are plugged to the main electricity supply and high-capacity Internet connections. Second, *sensors* that collect the information directly from the local application and also collaborate among themselves to forward the messages across the network to the destination. They are usually powered by batteries, so power consumption must be limited to extend lifetime of the devices.

JMAC includes a multi-hop scheme designed to minimize the energy consumption in sensor nodes. With this aim, the objective is keeping nodes sleeping as long as possible. They would only wake up periodically to receive a frame, and the rest of the time they can save energy by staying in stand-by/sleep mode. Thus, the underlying idea is to know when the next-hop node is awake to receive data and schedule the transmission of messages to that moment. To be able to predict next-hop’s waking-up time, each sensor must announce its time period *T* (the time while a node has its radio active for reception) and the remaining time (*Offset*) for its next receiving wake-up moment (*wakingTime*). That way, neighbours can estimate the right moment when the sensor will be ready to receive packets taking the arrival time of the last announce (*lastSeen*) and the transmission time of the message (*ToA*) into consideration as it is shown as follows:(1)wakingTime=lastSeen+Offset−ToA+n·T
where *n* is a multiplier to ensure that the estimated moment is in the future. This is the ideal approximation, but long transmission times on LoRa allows avoiding clock drifts and processing times on sensors and propagation delays of messages.

To enable upward routing, each sensor has to disseminate the distance to the gateway hopsToGateway to enable child sensors join the network and make parent nodes aware of their existence. The metric to route uplink messages is indeed the number of hops to the gateway hopsToGateway and the next upward hop selection is done dynamically from all those available direct parents (a node will create links with all the available nodes in range that are closer to the gateway), to choose the one which wakes up earlier. Additionally, downlink traffic from the gateway to sensors can be routed by recording in each node the list of children from each direct child. This way, parent nodes can extract from the frames identifiers of grandchildren nodes and schedule downlink messages to the right intermediate node to reach the final destination. Nevertheless, as downlink messages will be sporadic in data collection use cases, this version of the protocol does not include a downlink functionality, which will be completed as future work.

### 4.1. Devices Operation

Gateways behave as described by the Finite State Machine (FSM) shown in Figure 2a. Once the gateway is installed and connected to the backend (INIT), it is ready to execute an endless loop centralized in RECEIVE_UP_DATA mode to handle uplink messages from sensors or wait to send the next scheduled BEACON frame. Thus, if any message is received (frame), it will be used to update the topology information (neighbour_map). In case of an UP_DATA frame, it will also be processed to prevent duplication and an ACK frame will be sent to the sensor (SEND_ACK).

Otherwise, if the timeout for the next BEACON expires (beacon_timeout), the gateway switches to SEND_BEACON state to broadcast the BEACON and set beacon_timeout again to come back to RECEIVE_UP_DATA mode. BEACON frames are used for helping nodes to discover their neighbors. The time between each BEACON varies randomly between 15 and 25 s in order to avoid messages from a direct children to interfere always with the scheduled BEACON of the gateway, if there were the case. In this particular case, the first BEACON is sent (SEND_BEACON) right after initialization (INIT) to simplify development.

Sensors operate in two phases (Figure 2b): an initial one to discover the surrounding neighbours, and a second one which actually performs the JMAC protocol. The first one starts just after starting up, sensors need to wait in RECEIVE_INIT mode for a single frame which allows them to join the network. After that, they enter an announcing phase in which they announce periodically their information using a BEACON (SEND_BEACON) controlled by beacon_timeout, while the rest of the time they expect receiving announcements (frame) from their neighbours to update the known topology (RECEIVE_ANNOUNCE). The duration of this initialization phase can be considered to be negligible in consumption terms because the joining phase is set to a maximum of 5 min, and if it is not possible, the procedure is restarted and a notification is displayed through some user interface (e.g., a LED is turned on).

Upon completion of initialization phase, the new network is ready to run. First, each node opens a reception window to receive one UP_DATA
frame from a child (RECEIVE_DATA). If any frame is received, it is used to update neighbour_map and if it is also an UP_DATA
*frame*, the data payload is inserted in a queue and an ACK
frame is transmitted to the corresponding child (SEND_ACK). Conversely, if timeout expires or the received frame is not UP_DATA type, it directly goes to WAIT_OR_SLEEP mode.

In WAIT_OR_SLEEP state, sensors activate power-saving mode and estimate whether the closest parent (next_hop) will be listening on the channel during any moment of that time interval *T*. Additionally, a new parameter *C* is included which manages the probability of sending to the designated next_hop ($\frac{1}{C}$) in order to avoid potential collisions between sensors which would transmit at the same time. If all goes well, the sensor will wait until the desired next-hop’s awaking_time (WAIT_NEXT_HOP), otherwise until the sensor has to wake up again to start over the periodic operation (SLEEP).

At the end of the WAIT_NEXT_HOP state, the sensor checks if there is any pending data stored in the queues (both from the upper application layer and children nodes). If this is the case, a new UP_DATA frame is created with a packet from the local running application (if any) and at most $c_{max}$ aggregated packets from children. In other case it remains in sleep mode and changes to SLEEP state.

The generated UP_DATA frame is transmitted when the corresponding next_hop is expected to be awake (SEND_UP_DATA). If the packet is received correctly by the next_hop, the sensor will receive back an ACK and the information can be deleted from the queues (RECEIVE_ACK). After exiting the RECEIVE_ACK state, the sensor comes back to WAIT_OR_SLEEP mode. By doing so, it would be possible to send data to multiple parents in the same operational period *T*.

Finally, Figure 3 shows an example of the operation of the JMAC protocol when both kind of devices are involved (Gateway and Sensors).

### 4.2. Frame Format

The protocol operates using three types of frames that share the first two fields: (i) the *Type*, composed of 3 bits to distinguish the message type in reception (it has capacity for future messages types) and (ii) the *Source*, 8 bits long to identify the source node. Thus, it is possible to add new future messages types and, since the gateway is identified using the address 0, each network is limited to 255 sensors.

The first type of frame, *BEACON* (B), is used to advertise neighbours that a node exists in the coverage area. When sent by a gateway, it only carries the common part (blue part in Figure 4) because gateways are the sink nodes and are always listening on the channel (except when transmitting). When sent by sensors, they also include the green part in Figure 4), which is (i) the number of hops *Hops* to the gateway, which is only 2 bits and limits multi-hop to 4 hops; (ii) the time period *Period* and (iii) the time offset *Offset* with 64 bits to express each one to calculate the next awaking time accurately, and a list of at most $c_{max}$ child addresses.

The *UP_DATA* (U) frame (Figure 4) is generated by sensors to carry the application data of the current sensor and its children. Besides including all the fields that the *BEACON* (B) frame has, this second frame also contains a 10 bits sequence number *Sequence*, which is used to acknowledge the frame in a direct link and to associate the *My_Data* payload of fixed length of *M* bytes. It also contains the *Child_*_Data* payload and *Child_*_Sequence* sequence number to forward the information generated by its children nodes. If there is no application data in the current sensor, *My_Data* is empty.

Finally, the ACK (A) frame (Figure 5) confirms that the immediately preceding UP_DATA frame identified by Sequence_To_Ack was correctly received.

It is also worth mentioning that depending on the amount of data *M*, the corresponding network allows sending in the same UP_DATA frame up to $c_{max}$ children, considering the LoRa 255 bytes payload constraint. The way to calculate this limitation is expressed in Equation (Equation 2) and it limits UP_DATA and ACK frames to have a maximum length. Thus, for instance, when M=100 B, the network only supports 1 children, but when M=10B it is possible to have 18 children:(2)$$c_{max}=\left\lfloor \frac{255\cdot 8-3-8-2-64-64-10-M\cdot 8}{8+10+M\cdot 8}\right\rfloor$$

Both reception windows are controlled by a timeout set to the pertinent maximum frame length ($UP\_DATA_{max}$ or $ACK_{max}$) from a sensor node to allow completing the reception in the worst case scenario. Apart, the propagation delay is considered because nodes might be far away from each other and the propagation delay in LoRa is not marginal, so a distance of 15 km was chosen because it is the upper limit coverage in LoRa for urban areas.
(3)$$\begin{array}{c} timeout\left(UP\_DATA\right)=ToA\left(UP\_DATA_{max}\right)+propagationDelay\left(15\;\mathrm{km}\right) \end{array}$$
(4)$$\begin{array}{c} timeout\left(ACK\right)=ToA\left(ACK_{max}\right)+propagationDelay\left(15\;\mathrm{km}\right) \end{array}$$

### 4.3. Time Period

The time period *T* is the time when a node listens for new messages. In our protocol we take into account the worst case scenario (Figure 6) where a sensor node receives one UP_DATA frame and has to send back the corresponding ACK, and in average has the opportunity to transmit an UP_DATA frame to p/C parents, whose reception windows are inside the same time interval for the child sensor. Thus, the time period *T* is calculated respecting also the band duty cycle regulation. Precisely, this only affects the active usage of the channel, so as it expressed in the equation it only covers the transmissions of longest frames ($ACK_{max}$ and $UP\_DATA_{max}$) and is expanded according to the corresponding duty cycle DC limit.

In this way, when there are many available parents *p*, the time period *T* increases to ensure that sensors keep sleeping enough time to preserve low consumption. On the contrary, the higher parameter *C*, the lower time period *T*, which avoids having extremely long delays in final message delivery. The application payload length *M* has also little effect in its variation, as can be seen in Table 3.

## 5. Validation

We carried out a complete set of experiments using the OMNeT++ simulator [11] to validate our proposed protocol JMAC. We started by developing a realistic LoRa framework within OMNeT++ (Section 5.1). Then, we simulated a simple testbed to check the performance according to the different parameters (Section 5.2). Finally, we also studied the behaviour of JMAC using a more complex scenario with a denser topology to analyze the scalability and the performance of the protocol (Section 5.3).

### 5.1. FLoRaPHY: A New LoRa Framework for OMNeT++

LoRa is a radio technology with many features that are not currently supported in OMNeT++. Thus, we created a new module for OMNeT++ using FLoRa [30], which is focused on LoRaWAN, as a base for our work. FLoRaPHY is a new module that simulates the behaviour of the available information regarding the LoRa physical layer.

FLoRaPHY, follows the hierarchy of the INET framework [29], the OMNeT++ model suite for wired, wireless and mobile networks. This framework defines two big blocks: FlatRadioBase and RadioMedium. We extended them in (i) LoRaRadio and (ii) LoRaMedium, as Figure 7 details.

LoRaRadio extends the FlatRadioBase class and models the special LoRa reception issue of capturing the most powerful signal that is simultaneously available on the medium. This action does not truly reflect the actual behaviour of LoRa modulation because for capturing at most one of the signals the SNIR must be over a specific threshold depending on the SFs in each signal, but it facilitates the verification of the main protocol and assumes that collision rate is zero. It also computes the ToA of a frame according to the parameter setup and payload. It is composed of the following blocks (Figure 7a):LoRaTransmitter: it extends FlatTransmitterBase and is in charge of creating the transmissions.LoRaReceiver: it inherits from FlatReceiverBase and is responsible for computing whether signal decoding is possible according to the sensitivity of the transceiver and channel interference. It may discard the reception if it collides with interference signals as discussed above.IsotropicAntenna: it describes an ideal isotropic antenna, i.e., it radiates the same signal intensity in all directions.LoRaStateBasedEpEnergyConsumer: it records the power consumption of the radio module according to its state.

LoRaMedium decides which transmitted signals would arrive to each node according to the following models:LoRaAnalogModel: it models how a radio signal arrives to destination, specifically, it computes the final reception, RSSI and SNIR.LoRaPathLossOulu: it calculates the path loss of the radio signal inspired in the path loss model of the city of Oulu [64], but variability was reduced to simplify simulations.ConstantSpeedPropagation: it is used to emulate the propagation delay time of transmissions in accordance of traveled distance.IsotropicScalarBackgroundNoise: it adds uniform noise to the medium.

Additionally to this link layer, it is necessary to create the software to support the cross-layer protocol JMAC for OMNeT++, which combines the link layer (described above) and the network layer in a single one. With this aim, we created a new base class MacNetworkProtocolBase that combines both link and network layer functionalities. From it, we created the abstract class JMAC used to implement common features of the new protocol. This JMAC was extended for a particular gateway (JMacGateway) and sensor (JMacSensor).

Sensors were configured to wake up randomly when the simulation starts and they are fed from a dummy application layer which generates fake data of length *M* each 15 min from the moment they enter the operational loop. Finally, JMAC instances record information about the exchanged messages with the aim of generating a final report to evaluate our proposal according to the following three metrics:Packet delivery Ratio (PDR): it is the percentage of successfully received UP_DATA frames, i.e., UP_DATA frames which have received the corresponding ACK frame, over the total sent. It is local to the JMacSensor node and it usually converge if the system is static.End-to-end delay: it is the time taken for an application packet to arrive to the JMacGateway. It depends on the load of the path to the gateway, so it is a random variable which may be identified. If the system is congested it will grow infinitely because packets will never arrive to destination.Throughput: it is the ratio between the payload length of the application packet *M* and the end-to-end delay. It estimates the effective data rate which can be achieved by a sensor.

Finally, to adjust the simulator with realistic parameters, we used the characteristics of the SX1276 transceiver [65]. In order to estimate the average power consumption of each sensor and check if the theoretical consumption is compatible with a 1000 mAh battery, we modeled the consumption of the SX1276 transceiver in OMNeT++ according to the state of its radio: (i) when the transceiver is OFF, its consumption is 0.2A; (ii) if it is in SLEEP mode, its consumption is 1.5A; (iii) if it is RX_IDLE, the transceiver is ready but the channel is empty, its consumption is 1.6 mA; (iv) if it is RX, receiving a signal, its consumption is 10.3 mA; and, finally, (v) if it is TX, transmitting a frame, its consumption is 29 mA.

### 5.2. Testbed1: Checking the Impact of Parameters

In this first approach, we used a simple and low-density network (Figure 8a) and a small combination of possible values for the parameters (M=30 B, 50 B, 100 B and C=1,2,3,4). The three aforementioned metrics (PDR, end-to-end delay and throughput) were evaluated on average over 20 runs of each experiment, which emulates the operation of the network during 7 days.

According to the results summarized in Figure 9a, we can conclude that using a higher *C* parameter significantly improves the PDR in all sensors, since it makes less probable to have two or more devices transmitting during the same receiving window. Thus, a low value for *C* could cause interference and all devices, except the most powerful one, would have to retransmit. The application payload length *M* has very little impact, except for the particular case M=100 and C=1 because it only allows sending information about one child at a time causing high load periods that will generate more messages and potential collisions.

It is important to remark that:Sensors 1 and 2 achieve similar results, but even when *C* is higher than one the PDR is not 100 % because the gateway does not have a specific receiving window and it may happen that UP_DATA messages interfere with BEACON or ACK frames from the gateway, but still performance is really good.Sensor 4 performs worse than sensors 3 and 5 because both parents (sensors 1 and 2) are shared with the other sensors, and there are more probability to interfere.Sensor 6 has better PDR compared to sensors 7 and 8 because it has his own parent (sensor 4), apart from the shared one (sensor 5).The PDR in sensor 10 is quite better than its sibling sensor 9 because it benefits from two parents simultaneously.

Regarding energy consumption, for the selected values for *M* and taking into account that time period *T* decreases when they increase, we can observe how sensors stay active more time. Then, as shown in Figure 9b, they tend to waste more energy when a higher payload is supported. Contrary to what would be expected, power consumption increases with parameter *C* because time period *T* is smaller and sensors keep sleeping less time. However, it increases lightly because the higher *C* the more packets are grouped in a single frame, when possible. Besides, some sensors suffer from abnormal high power consumption for M=100 and C=1, this peak results from the successive retransmissions which have to be done when the frame is interfered by another one. This is the case for sensors 1, 2 4, 5, 6, 7 and 8, which have a low PDR in that situation.

Figure 10 shows the results obtained for both average end-to-end delay and throughput. It can be concluded that the higher the number of hops to the gateway, the higher delay and the lower throughput. Concretely, delay and throughput are proportional and to the time period *T* and the distance to the gateway. Moreover, *C* parameter has a noticeable impact in both metrics. In average, C−1 out of *C* reception windows are skipped, so an increase in the value of *C* means further delay and slower effective data rate. However, this effect is compensated with time period *T* formula, which decreases with higher *C*. Furthermore, application payload *M* only has remarkable impact on the delay of sensors which have several children. That is the case for M=100, which only supports sending information about a single children in each UP_DATA frame and latency is increased because messages keep in the waiting queues longer when the network traffic is substantial. However, the impact of the payload length *M* is not marginal for the throughput and it will increase proportionally the effective data rate. There is a large variance anomaly of the delay for M=100 and C=1 for sensors 7, 8 and 10 because they share parents 5 and 2, which are nodes with many possible end-to-end paths. The effect is less severe in sensor 10 because it also has sensor 6 as parent.

### 5.3. Testbed2: Checking the Scalability and Performance

A second scenario (Figure 8b) was used to study the scalability and performance of the protocol. Specifically, it was configured with M=30 B, because most of the target applications for this protocol only require transmitting small amounts of data, and we used C=3 in order to achieve a good PDR and not increasing delay and consumption. The simulation replicates a scenario executing the protocol for 7 days and repeating 20 times, so the following results depict the average behaviour.

Considering the average results from Figure 11a, the attained PDR is generally higher for situations with low traffic demand. One outstanding case is the good performance of sensors 16 and 17 (higher than 99%). This is because they have three common parents, which allows them to balance the traffic. As shown in Figure 11c,d, end-to-end delay increases and throughput decreases with the number of hops to the gateway, but they are normally within reasonable values for this scenario. Having several parents imposes a substantial increment in the final delay, such is the case of sensors 17 and 18, which considerably reduce the data rate. Finally, and regarding the power consumption (Figure 11b), they turn out as expected. On one side, leaf nodes consume a low amount of power as they rarely transmit and do not usually receive a message for them. On the other side, sensors which act as relayers require higher amounts of energy.

Given this realistic scenario, it is useful to estimate whether an ordinary battery (1000 mAh) is adequate for having long life cycle. To do so, the average highest consumption will be selected (sensor 1) and a power supply of 3.3 V will be assumed. The estimated duration of the battery for the worst case under ideal circumstances assumed in this study is longer than one year, so this deployment could be adopted without the need for maintenance for a long period of time:$$LifeCycle=BatteryCapacity\cdot \frac{PowerSupply}{PowerConsumption}=1000\;\mathrm{mAh}\cdot \frac{3.3\;V}{239.6\;\mathrm{\nu W}}=13,772.96\;h.\approx 573\mathrm{days}$$

## 6. Discussion

After analysing the conducted experiments, we can conclude some relevant aspects. First, the *C* parameter always cause a notable variation in the performance: the higher *C*, the better PDR, but end-to-end delay, throughput and consumption tend to worsen. Second, the value of *C* has a major impact when the system is heavily charged, because it allows aggregating more application packets in a single frame and interference between signals is less probable.

Besides, the application payload length *M* only affects to the performance of a sensor when it has more children than the maximum supported $c_{max}$ or the paths to the gateway are shared between many siblings. Additionally, a higher number of available parents for a sensor increases the PDR, but also the latency. Having a much denser network may enable more paths for children sensors, which increases scalability of the solution.

During the protocol design and evaluation, some insights were gained to modify and enhance the proposal. Regarding the infrastructure deployment, it is highly recommended to use RSSI and SNR measurements to set up routing links between nodes. This is because in urban areas signals are scattered and there may be lossy links, but it was not included in this version because it would be necessary to develop a new model for the channel degradation effects (interference, noise, obstacles...). Besides, we suggest to adjust the *C* parameter locally by each sensor to pseudo-coordinate the transmissions to avoid possible collisions. Ideally, it would be the total number of sibling sensors, but sensors may not know all their neighbours.

Regarding the protocol behaviour, there are some aspects that might be improved or, at least, checked to assess their impact. First, it could be convenient for sensors to include the transmission of a BEACON at the beginning of each time interval, because there may be occasions when a sensor does not send any frame with information and could delay the joining procedure of a new sensor. This extra BEACON would only be sent if there is no UP_DATA frame scheduled to send in that time interval, and could be used to ensure that a sensor is still active and improve mobility aspects. Another improvement for reducing power consumption is to force sensors located near the gateway to send UP_DATA frames only when their transmission buffer is full. Nevertheless, this would increase the end-to-end delay, so it is necessary to reach an equilibrium. It would also be interesting to check the performance with a unique queue for pending messages, instead of using two queues which may prioritize the messages from the local sensor.

Finally, it is worth comparing the proposed protocol with the most relevant and similar proposals about LoRa multi-hop exhibited in Section 3. Table 4 shows the most relevant features considered for the design of the different protocols. As shown, JMAC deals with almost all critical points, but further work must be done to enable auto-configuration of nodes and facilitate deployments. Besides, it addresses energy consumption and keeps control over duty cycle of sensors, which are important issues in IoT solutions that are not usually covered.

## 7. Conclusions and Further Work

LPWAN technologies are designed to support long range communications with low power consumption. However, most of them are based on single hop communications, which may entail problems in areas where the coverage range is reduced because of contextual elements (interference, noise, obstacles, etc.). This is especially relevant for indoor locations (such as in industrial settings) or in crowded areas (such as smart cities). Our JMAC proposal, a cross-layer multi-hop protocol for LoRa, was conceived to overcome this issue by combining two successful strategies. On the one hand the LoRa radio technology to keep low energy consumption and extend the coverage area and, on the other hand, enabling multi-hop capabilities to reuse resources and expand the coverage area without adding more gateways, which usually make the infrastructure more expensive. Our proposal is also compliant with the main general requirements of LPWAN [1], with the exception of using ALOHA with single-hop routing, as this was precisely the reason for this research work. This first contribution is complemented with a new simulation framework within the OMNeT++ simulator, coined as FLoRaPHY, to simulate as close as possible to the real behaviour of LoRa to check the operation and scalability of the new protocol JMAC. Finally, and according to the simulation results, we can conclude that our proposal ensures a very low power consumption, facilitating the deployment it in real scenarios with sensor powered by batteries. It is interesting to mention that our approach can supplement the current LoRaWAN protocol, offering a more powerful strategy to increase the coverage.

There are some open issues that we plan to study in a future work, such as a theoretical study of the performance of the JMAC protocol. It requires a deeper understanding of the physical modulation of LoRa and the formulation of a mathematical model for LoRa signal collision. This way, all other metrics, such as end-to-end delay, throughput, error rate, etc., could be better-modelled. Besides, more experiments can be performed to estimate the maximum capacity of the network. One option is to allow sensors to generate data more often and study when the network becomes saturated and how it thereby performs to set a maximum time to generate application data. LoRa allows simultaneous virtual channels (combination of SF and frequency channel) to operate without interfering between them, so it would be possible to increase the total capacity of the deployment by creating different multi-hop networks allocated to different virtual channels. This idea has already been explored in [66] to improve scalability. It would also be interesting to adapt the protocol to the existing LoRaWAN solution, achieving compatibility, so they can be better compared in terms of performance. Of course, security aspects must be also considered a critical priority in future work for multi-hop strategies. Last, but not least, we are currently working on an implementation of the solution in a real world setting to assess the performance and to elaborate a new propagation loss model for urban areas as well as to explore the actual coverage range of the solution. Within this context, we are also working on the downlink flow, in order to have a complete solution for the JMAC protocol.

## Figures and Tables

**Figure 1 sensors-20-06893-f001:**
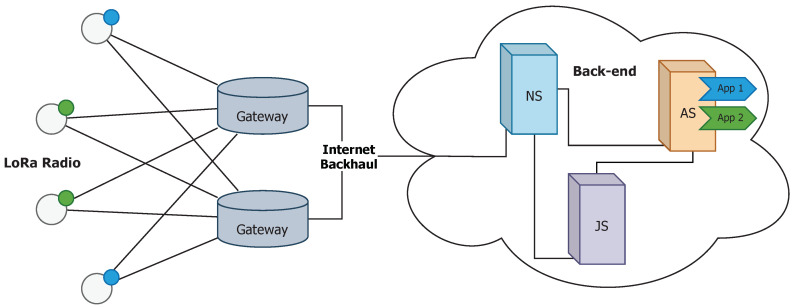
LoRaWAN architecture.

**Figure 2 sensors-20-06893-f002:**
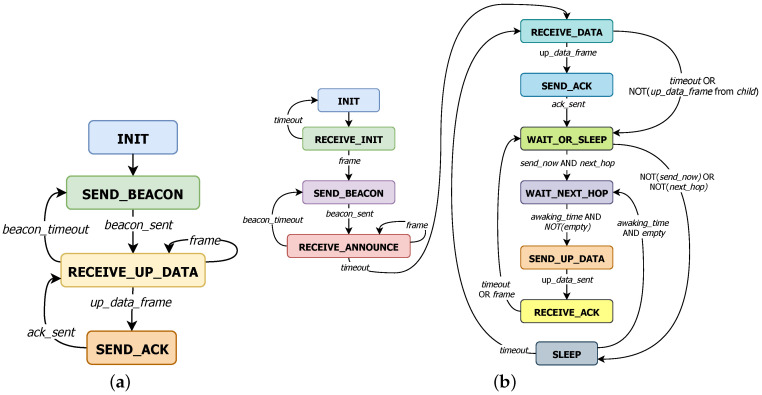
Finite State Machine (**a**) Gateway (**b**) sensor.

**Figure 3 sensors-20-06893-f003:**
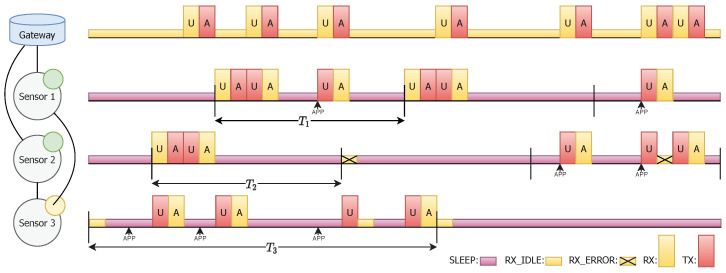
Example of the operation of the JMAC protocol (U → UP_DATA, A → ACK).

**Figure 4 sensors-20-06893-f004:**
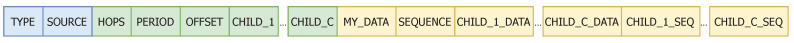
Format of the *UP_DATA* frame from sensor.

**Figure 5 sensors-20-06893-f005:**
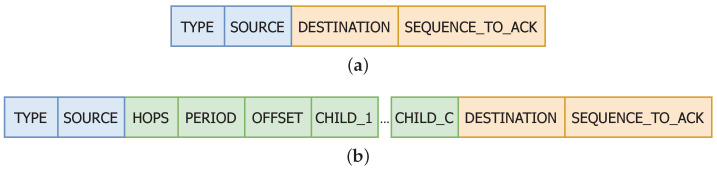
Format of (**a**) the ACK frame from gateway and (**b**) the ACK frame from sensor.

**Figure 6 sensors-20-06893-f006:**
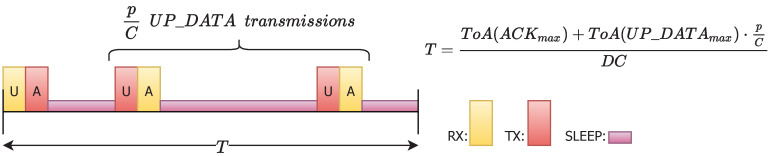
Average worst case situation of a time period in a sensor.

**Figure 7 sensors-20-06893-f007:**
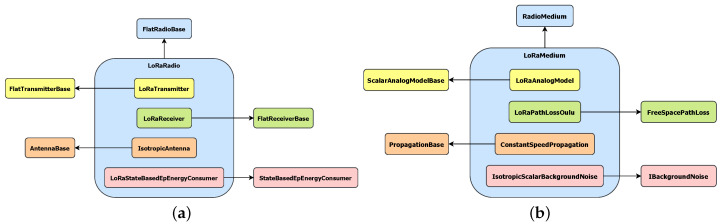
Structure of (**a**) LoRaRadio class and (**b**) LoRaMedium class.

**Figure 8 sensors-20-06893-f008:**
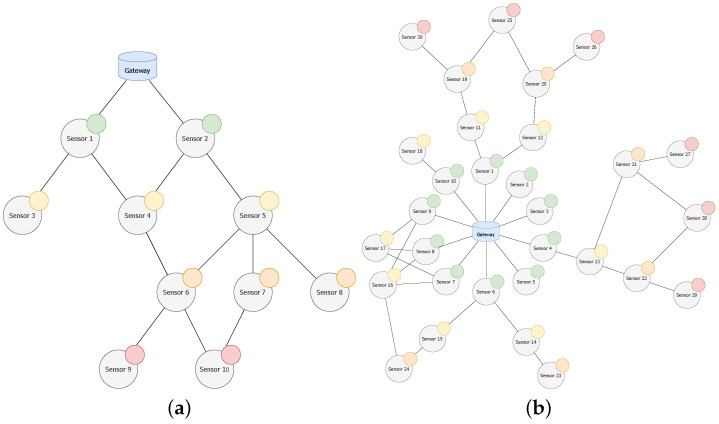
Topology for (**a**) testbed 1 and (**b**) testbed 2.

**Figure 9 sensors-20-06893-f009:**
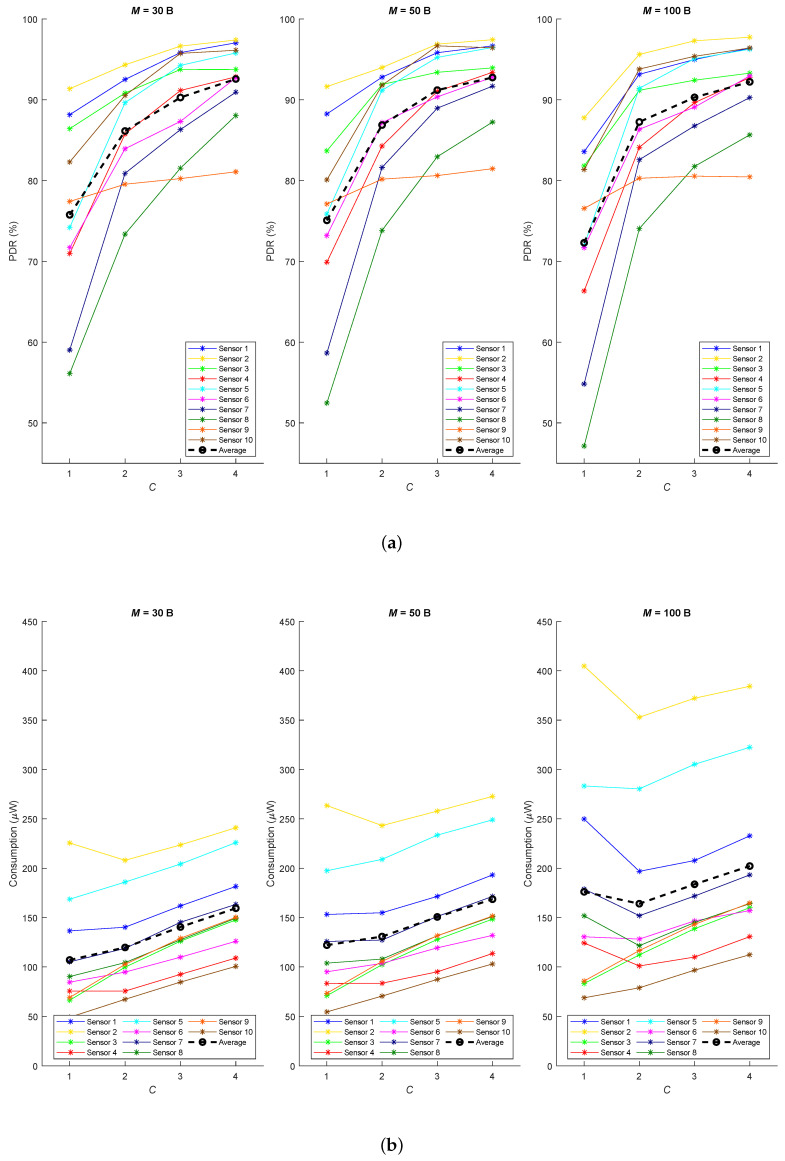
Results for the Testbed1 (**a**) PDR and (**b**) power consumption.

**Figure 10 sensors-20-06893-f010:**
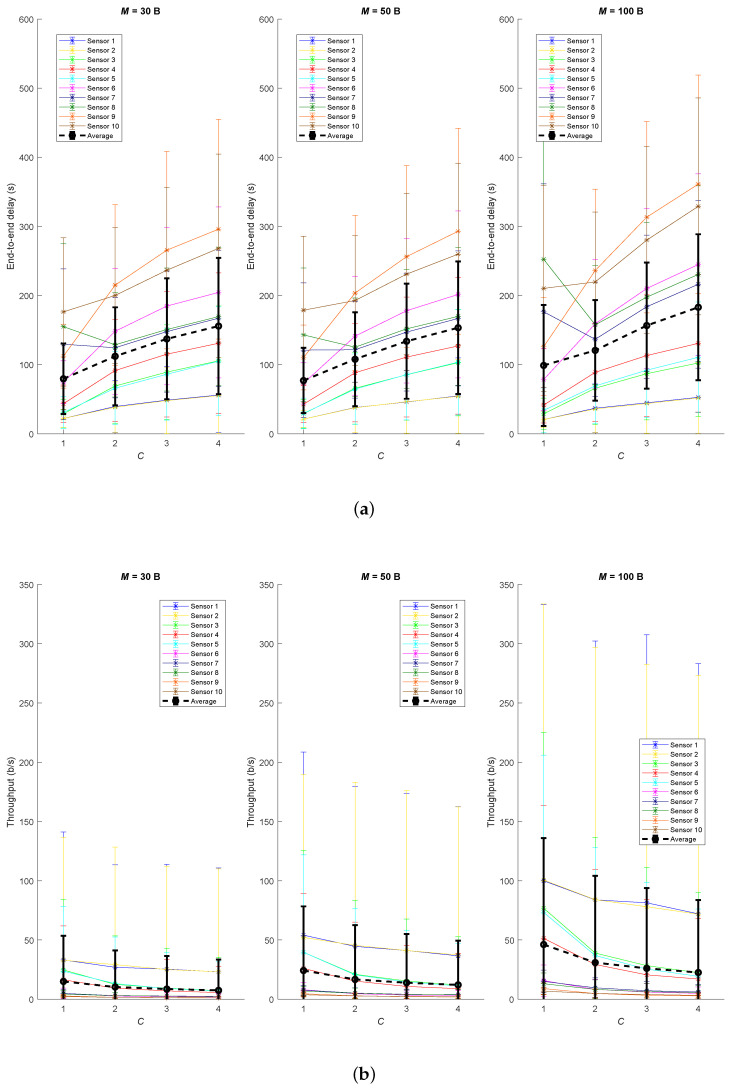
Results for the Testbed1 (**a**) average end-to-end delay and (**b**) throughput.

**Figure 11 sensors-20-06893-f011:**
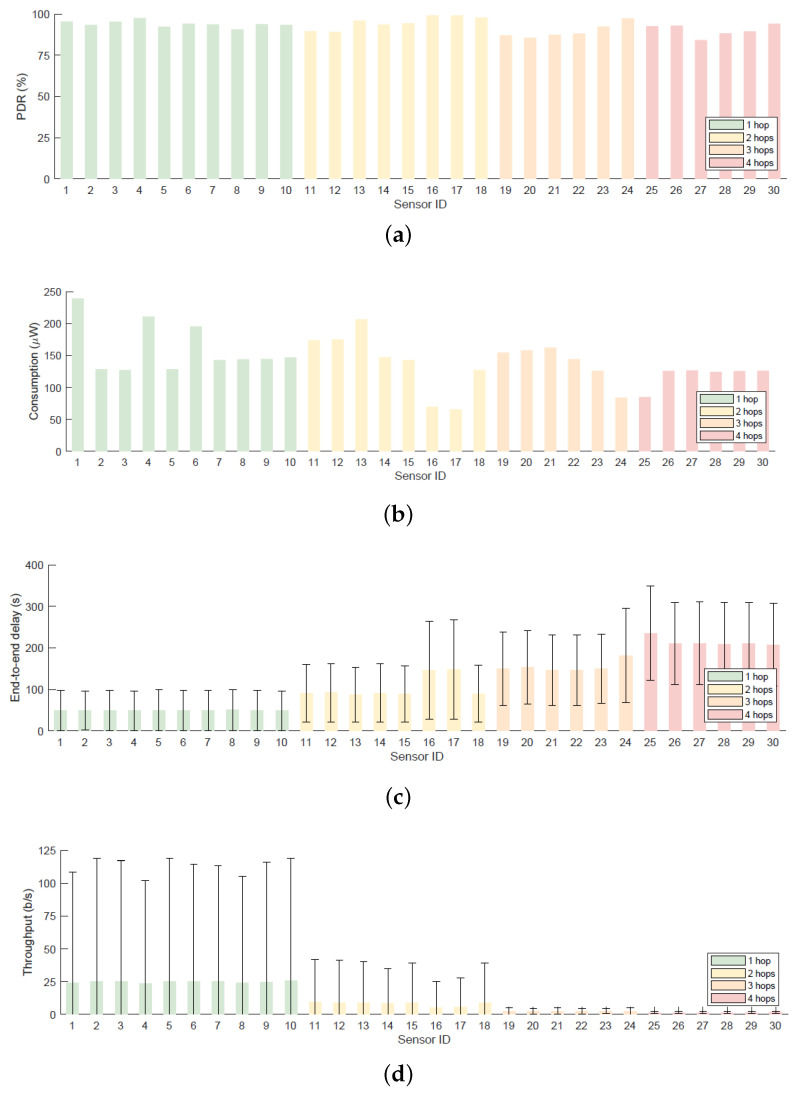
Results for the Testbed2 (**a**) PDR, (**b**) power consumption, (**c**) average end-to-end delay and (**b**) throughput.

**Table 1 sensors-20-06893-t001:** Summary of the structure, main objectives and evaluation of the LoRa/LoRaWAN proposals.

Ref.	Link Layer	Network Layer	Cross-Layer	Objective	Reliability/ Consumption/ Latency	Duty Cycle Limit
[33]	CT TDMA	Broadcast	No	Reliability	High/-/-	No
[34] [35]	Centralized	Tree Distance Vector	No	Reliability	High/High/Medium	No
[36]	TDMA CAD	Tree Distance Vector	Yes	Consumption Reliability Latency	High/-/Medium	No
[37]	-	Anycast with memory	No	Reliability Latency	Medium/High/Low	No
[38]	TDMA	Broadcast	No	Consumption Reliability	High/Medium/Low	No
[39]	Distributed TDMA and CAD	Tree Distance Vector	Yes	Reliability Latency	High/-/Low	No
[41]	RLMAC	RPL	No	-	-/-/Medium	Yes
[42]	-	RPL	No	-	-/-/-	Yes
[43]	TDMA CSMA/CA	RPL	No	-	-/-/High	Yes
[45]	TSCH	RPL	No	Reliability	-/-/-	Yes
[46]	LBS	IPv6	No	Reliability Security	High/-/-	No
[48]	CAD	Modified LoRaWAN and DSDV	No	Reliability	Medium/-/-	No
[49]	TDMA	Tree Distance Vector	No	Reliability	Medium/-/Medium	No
[50]	TDMA	Tree Distance Vector	Yes	Reliability	-/-/-	No
[51]	CAD	HWMP AODV	No	Reliability	-/High/Low	No
[52]	Modified LoRaWAN	AODV	No	Reliability	-/High/Low	No

**Table 2 sensors-20-06893-t002:** Summary of the structure, main objectives and evaluation of the WSN protocols.

Reference	Link Layer	Objective	Reliability/ Consumption/ Latency	Duty Cycle Limit
[53]	RTS/CTS	Consumption	-/Medium/Medium	No
[54]	RTS/CTS and Early Timeout	Consumption	-/Low/Medium	No
[55]	RTS/CTS and Combining Packets	Consumption	-/Low/Medium	No
[56]	Distributed TDMA	Consumption	-/Low/Low	No
[57]	Preamble Sampling and Neighbour Learning	Consumption	Medium/Low/Medium	No
[58]	CCA and Preamble Sampling	Consumption	High/Low/Medium	No
[59]	Sleep-Awake Pattern and Neighbour Learning	Consumption Throughput	High/Medium/-	No
[60]	Strobed Preamble Sampling	Consumption	Medium/Low/Medium	No
[61]	Asynchronous Wake-up and Neighbour Learning	Consumption Reliability	High/Low/Medium	No
[62]	Target Preamble Sampling	Consumption Reliability	High/Low/Low	No
[63]	Target Preamble Sampling and Pseudo-Random Neighbour Learning	Consumption Reliability	High/Very Low/Low	No

**Table 3 sensors-20-06893-t003:** Time period *T* for an application payload of (a) M=30 B and (b) M=100 B.

M=30 B	p=1	p=2	p=3
C=1	44.032 s.	81.9968 s.	119.9104 s.
C=2	25.1264 s.	44.0832 s.	63.0400 s.
C=3	18.8075 s.	31.4453 s.	44.0832 s.
C=4	15.6480 s.	25.1264 s.	34.6048 s.
M=100 B	p=1	p=2	p=3
C=1	40.4992 s.	75.3508 s.	110.1824 s.
C=2	23.0784 s.	40.4992 s.	57.9200 s.
C=3	17.2715 s.	28.8853 s.	40.4992 s.
C=4	14.3680 s.	23.0784 s.	31.7888 s.

**Table 4 sensors-20-06893-t004:** Comparison of addressed topics in JMAC and other releveant LoRa multi-hop protocols.

Proposal	Coverage	Consumption	Latency	Throughput	Routing	DutyCycle Control	Auto-Configuration
JMAC	x	x	x	x	x	x	
[33]	x		x		x		
[34][35]	x		x		x		x
[36]	x	x	x		x		
[38]	x	x	x		x		x
[39]			x		x		x
[48]	x			x	x		x
[49]	x	x	x	x	x		x
[50]	x				x

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
