# Peer review of "JMAC Protocol: A Cross-Layer Multi-Hop Protocol for LoRa"

_sensors, 2020, doi:10.3390/s20236893_

Round 1

Reviewer 1 Report

The paper presents a new multi-hop protocol, called JMAC, based on LoRa radio technology, designed with the aim of extending coverage and the number of nodes supported by each cell in monitoring scenarios such as smart cities and industry 4.0.

The paper is very well structured, detailed and written. It includes an extended and up-to-date set of references and just a few minor issues came up from reviewing process:

#1 – background and related work sections and summary tables 1 and 2 fit very well on the paper. However, nothing is said about the criteria used to identify Reliability, Consumption and Latency in both tables.

#2 – in section 4, line 329, authors wrote “...we consider they are rarely sent, so they must be prioritized…”. As this paper deals with a multi-hop protocol (messages), in my opinion, it is important to clearly state, here, that down-link messages are not considered in this protocol proposal/version.

#3 – Lines 511, 512, 513. It seems that, at least, units are wrong. Authors should verify consumption values at least for RX and TX operations. They seem too much low.

#4 Line 531, authors wrote “...Sensor 4 performs worse than sensors 3 and 4 because…”. Probably authors want to compare sensor 4 with sensors 3 and 10 ?!

#5 – Discussion section seems very interesting but lacks the comparison with other proposals. It would be very useful for the reader if authors provided a comparison with the works (or some) referred in the related work section. If comparison is not possible, for any reason, authors should also state that.

#6 – Two typos (?): degree symbol above workingtime formula and in line 342 there is "Km." (with or without the dot?)

Reviewer 2 Report

In this paper, the authors presented a new protocol called JMAC, as a cross-layer multi-hop protocol applicable for long range low-power wide area networks. As per the paper, this is new multi-hop protocol designed for improving long range wireless communication networks. They also claim that, this protocol is designed to minimize energy consumption in sensor nodes.
Overall this is a well-written paper, and authors presented the concepts clearly with details, and also described the frame format, and theory behind designing this protocol. However, there is a scope for improvement. Here are some comments for improving the manuscript.

  1. One may argue that, this fundamentally a multi-hop protocol, with links to each nodes, and forms a tree-structure. In this connection, as shown in Fig. 4, the frame format shows that, there are fields for several CHILD_X. Does this means, that, the frame must included all the information of child of a node or connected nodes? More details are required.
  2. Is there the limit on number of links to a child node or neighbor nodes? If so how this is determined? For example, as shown in Fig. 3,each nodes seems having two links to connect other nodes.
  3. As per the information presented in this paper, the authors introduced this protocol using theoretical framework and implemented using simulator (OMNeT++). But in practice, it is only after implementing a sample physical test bed, one is able to evaluate the true potential, working mechanism, limitations, and performance of the new protocol, especially when it comes of energy consumption, and long range wireless communication. More information is necessary.
  4. If this protocol to be implemented in a physical test bed, what kind of sensors can be used in this context? How to perform such testing especially for evaluating energy consumption. Whether one can depend on LoRa modules only? More details are expected.
  5. There is a symbol degree at the end of passage starting with line 344, after the word ...as follows. I guess it is not required.
